# Independent Evolution Has Led to Distinct Genomic Signatures in Dutch Elm Disease-Causing Fungi and Other Vascular Wilts-Causing Fungal Pathogens

**DOI:** 10.3390/jof9010002

**Published:** 2022-12-20

**Authors:** Anna Fijarczyk, Louis Bernier, Monique L. Sakalidis, Carmen M. Medina-Mora, Ilga Porth

**Affiliations:** 1Département de Biologie, Université Laval, Québec, QC G1V 0A6, Canada; 2Institut de Biologie Intégrative et Des Systèmes (IBIS), Université Laval, Québec, QC G1V 0A6, Canada; 3Département des Sciences du Bois et de la Forêt, Université Laval, Québec, QC G1V 0A6, Canada; 4Centre d’Étude de la Forêt (CEF), Université Laval, Québec, QC G1V 0A6, Canada; 5Department of Plant, Soil and Microbial Sciences, Michigan State University, East Lansing, MI 48824, USA; 6Department of Forestry, Michigan State University, East Lansing, MI 48824, USA

**Keywords:** Dutch elm disease, *Ophiostoma*, vascular wilt diseases, comparative genomics

## Abstract

Vascular wilts are important diseases caused by plant pathogenic fungi that result in the rapid death of their plant hosts. This is due to a systemic defense mechanism whereby the plant induces the compartmentalization of the infected vascular system in order to reduce the propagation of the fungus. The ascomycete class Sordariomycetes contains several species that cause vascular wilts in diverse plant hosts, and they can be classified into four taxonomic orders. The genetic mechanisms of pathogenesis have already been investigated in *Fusarium* and *Verticillium* species, but they have not yet been compared with other well-known wilt-causing species, especially fungi causing oak wilt or Dutch elm disease (DED). Here we analyzed 20 whole genome assemblies of wilt-causing fungi together with 56 other species using phylogenetic approaches to trace expansions and contractions of orthologous gene families and gene classes related to pathogenicity. We found that the wilt-causing pathogens evolved seven times, experiencing the largest fold changes in different classes of genes almost every time. However, some similarities exist across groups of wilt pathogens, particularly in Microascales and Ophiostomatales, and these include the common gains and losses of genes that make up secondary metabolite clusters (SMC). DED pathogens do not experience large-scale gene expansions, with most of the gene classes, except for some SMC families, reducing in number. We also found that gene family expansions in the most recent common ancestors of wilt pathogen groups are enriched for carbohydrate metabolic processes. Our study shows that wilt-causing species evolve primarily through distinct changes in their repertoires of pathogenicity-related genes and that there is the potential importance of carbohydrate metabolism genes for regulating osmosis in those pathogens that penetrate the plant vascular system.

## 1. Introduction

The fungal pathogens that attack trees represent a heterogeneous and widely diverse assemblage in terms of phylogeny, ecology, and physiology. All tissues and organs produced by trees can be colonized by those pathogens. Some species develop over small and often well-defined portions of tissue and produce discrete symptoms, such as leaf spots. Others spread rapidly through large portions of the host tissue and induce systemic damage, visible as blackened streaking in the vascular tissue and wilting and premature defoliation of leaves. The latter occurs as a result of the invasive growth of aggressive pathogens in the xylem vessels of hardwoods and tracheids of conifers. Vascular wilts are also known in herbaceous plants and are mostly caused by *Fusarium* and *Verticillium*. However, the coevolution of pathogenic fungi with trees has been traced back already to Jurassic conifers, for which fossil evidence suggests that the formation of tyloses was accompanied by fungal colonization in their tracheids [1]. Among trees, wilt diseases are particularly known in ring-porous species of hardwoods that depend on their xylem sap conductive system of large vessels for their high hydraulic conductivity, especially at the start of each growing season.

The host plant can limit the propagation of the fungal hyphae by blockage of vessel elements through the formation of structures such as tyloses that emerge as outgrowths from vessel adjacent parenchyma cells reaching into the lumen of infected xylem vessels. We also note that tyloses are also produced in response to drought. Sealing off affected vessels occurs by the deposition of pectin material around these tyloses structures, also initiated by parenchyma cells, which helps to compartmentalize the vascular fungus [2,3]. Reduced susceptibility to wilt infections is found in tree species that already have their vessels partially occluded naturally. For example, different sensitivities to *Bretziella fagacearum* (the oak wilt-causing fungal pathogen) are known among *Quercus* species belonging to different sections within the genus *Quercus* in North America, with Lobatae being the most affected section with mostly ring-porous species. Other forms of tree defenses have also been reported against *Ophiostoma novo-ulmi* (a Dutch elm disease (DED)-causing fungal pathogen), involving the induced deposition of ligno-suberin in xylem tissue of *Ulmus* [4,5]. Suberin is thought to act by reinforcing vascular cell walls against degradation by *O. novo-ulmi,* and by producing a physical-chemical barrier to restrict fungal propagation in the xylem [6]. This process has also been reported in plant species other than trees [7].

Infection by vascular wilt disease fungi can have catastrophic consequences for the plant host, such as rapid death of the host and spread of the disease to other adjacent hosts. Historically, DED has decimated natural and planted elm populations worldwide since it was first reported in 1922 [8]. The two successive DED pandemics that have been documented [9,10,11] were caused by two species belonging to the order Ophiostomatales, the moderately aggressive *Ophiostoma ulmi* and the highly aggressive *O. novo-ulmi* [12]. Oak wilt, caused by *Bretziella fagacearum* (Microascales), is another well-studied vascular wilt [13]. It affects mostly the members of the red oak group (sect. *Erythrobalanus*) within the genus *Quercus* and has been expanding steadily throughout the United States during the last decades. The pathogens responsible for DED and oak wilt are vectored by insects and spread underground from tree to tree via interconnected root systems and, under natural conditions, affect tree species from only a few genera. In contrast, *Verticillium ahlia* and *V. albo-atrum* (Glomerellales), the aetiological agents of Verticillium wilt, are soilborne pathogens that infect their host through the roots. Verticillium wilt has been recorded on numerous plant species, including over 100 taxa of trees and shrubs [14]. The best-known vascular wilt of conifers is the black stain root disease [15] caused by *Leptographium wageneri* (Ophiostomatales). Like for *O. ulmi*, *O. novo-ulmi* and *B. fagacearum*, the pathogen is vectored by insects, namely, root- and root collar-feeding bark beetles and weevils in the genera *Hylastes*, *Hylurgops*, and *Pissodes* [16].

The wilt pathogens mentioned above, along with several others, have been studied for over 50 years. New emerging wilt pathogens have joined the list of biological agents threatening the survival of various tree species. For instance, *Raffaelea lauricola* (recently renamed to *Harringtonia lauricola* [17], Ophiostomatales), the causal agent of laurel wilt, which was first recorded in 2003 [18,19], has since killed hundreds of millions of redbay (*Persea borbonia*) trees in the southeastern Atlantic Coastal Plain region of the United States [20]. In Hawai‘i, a vascular wilt disease known as Rapid ‘Ōhi‘a Death has been causing unprecedented mortality of ‘Ōhi‘a trees (*Metrosideros* spp.) since 2012. Molecular analyses and verification of Koch’s postulates have shown that two novel species in the Microascales, the highly aggressive *Ceratocystis lukuohia*, and the less aggressive *C. huliohia*, were responsible for this disease, which could potentially result in catastrophic losses in ecosystem services [21]. These events illustrate the need for further investigations of the evolutionary history, biology, and genomics of wilt pathogens in order to design long-term management and mitigation strategies.

Because of the economic importance of vascular wilts of agricultural crops, most of our current knowledge of the biology and genomics of wilt fungi was gained from studying *V. dahliae*, *V. albo-atrum*, as well as *Fusarium oxysporum forma specialis lycopersici* (Hypocreales), which is the causal agent of Fusarium wilt. Since the plant xylem is a nutritionally poor environment, vascular pathogens rely on (i) efficient mechanisms for acquiring and processing nutrients available in the xylem sap; (ii) enzymatic digestion of host cell walls; (iii) invasion of neighboring cells; or (iv) induction of nutrient leakage from surrounding tissues [2]. Among these strategies, enzymatic digestion of host cell walls is the one for which there is more compelling biological and physiological evidence across taxa of fungal wilt pathogens, including the DED-causing fungi *O. ulmi* and *O. novo-ulmi* [22,23,24,25].

In their paper comparing the genome of *F. oxysporum* f. sp. *lycopersici* (*FOL*) with those of the non-wilt-causing species *F. graminearum* and *F. verticillioides,* Ma and colleagues [26] associated six genes located on the lineage-specific chromosome 14 of *FOL* with the ability of the latter to cause wilt. Klosterman and colleagues [27] later compared the genomes of *V. albo-atrum*, *V. dahliae,* and *FOL* to those of seven other fungi that were either non-wilt pathogens or saprobes. They pointed out that the genomes of the two *Verticillium* species contained significantly higher numbers of genes encoding carbohydrate-active enzymes (CAZymes) belonging to the polysaccharide lyases (PL) family. Moreover, these authors identified 10 predicted proteins that were shared by the three wilt pathogens but were absent in the remaining seven species. None of the genes encoding these predicted proteins [27] seemed homologous to any of the six genes associated previously with pathogenicity in *FOL* [26]. One of the proteins, annotated as a glucosyltransferase, appeared to have been acquired by the horizontal transfer from a bacterial ancestor. Deletion of the glucosyltransferase gene in *V. dahliae* did not affect pathogenicity towards lettuce (*Lactuca sativa*) but resulted in reduced virulence towards benthi (*Nicotiana benthamiana*) [27].

The above-mentioned comparative studies, which have been cited abundantly over the last 10 years, were conducted at a time when very few complete fungal genomes were available. Since then, continuous technological improvements and diminishing sequencing costs have led to the availability of an increasing number of fungal genomes. Yet, to our knowledge, there has been no renewed effort to conduct genome-wide analysis in order to identify genomic signatures that may be specific to vascular wilt pathogens, including agriculturally important pathogens and non-model species such as the DED-causing fungi and their Ophiostomatoid relatives. We, therefore, searched public databases for relevant genomes and, in the case of Ophiostomatales, included the additional genomes sequenced through the bioSAFE project [28,29]. This resulted in a database consisting of genomes from 76 species with contrasting lifestyles (vascular wilt pathogen, non-wilt pathogen, saprobe) belonging to the orders Glomerellales, Hypocreales, Microascales, and Ophiostomatales within the Sordariomycetes. Our comparative analysis of the gene families and classes suggested that the ability to cause vascular wilt had evolved independently in pathogens belonging to phylogenetically different groups.

## 2. Materials and Methods

### 2.1. Wilt-Causing Species and Genome Assemblies

We downloaded genome assemblies of 20 strains of Ascomycetes from orders Hypocreales (*n* = 3), Glomerellales (*n* = 5), Microascales (*n* = 5), and Ophiostomatales (*n* = 7) that are known to cause vascular wilt, as well as 56 other pathogens and saprotrophic species belonging to the same or different Sordariomycetes orders for comparison (Appendix A). The wilt-causing species included species that colonize host xylem systemically, except for two members of the Ophiostomatales, *Raffaelea quercivora,* and *Leptographium procerum*, which develop less extensively in infected vessels and usually need to be inoculated at higher densities in order to kill their host [30,31]. Genome assemblies were downloaded from NCBI (*n* = 61), JGI Mycocosm database (*n* = 11) [26,27,32,33,34,35,36,37,38,39], and other resources (*n* = 1) or were sequenced previously by the authors (*n* = 3) [40]. The *Bretziella fagacearum* C519 v1.0 genome was sequenced and assembled as part of the 1000 genomes project and is available in JGI Mycocosm. All genome assemblies were processed as described in Fijarczyk et al. [40]. Short contigs were excluded (<1000 bp) and contaminated contigs were removed. Contaminants were identified by BLAST searches of contigs separately against bacterial/viral UniProt accessions and against fungal UniProt accessions, after which, those contigs with more bacterial/viral hits in length than fungal hits were removed. The completeness of the assemblies was evaluated by searching for conserved single-copy genes from the Sordariomycetes fungal group with Busco v3 [41].

### 2.2. Gene Molecular Functions

All genomes were searched for ab initio gene models with Augustus v.3.3.2 [42] using training sets from *Botrytis cinerea*, *Fusarium*, *Magnaporthe grisea*, *Neurospora,* or *Verticillium longisporum*. The species training file for *Ophiostoma novo-ulmi, O. ulmi, O. himal-ulmi,* and *O. quercus* was generated as described in Fijarczyk et al. [40]. Orthologs and orthogroups were identified with Orthofinder v2.5.2 [43], where no a priori tree was defined. Orthogroups comprise all gene homologs within and between species and can have one or more members per species. Species gene models (protein sequences), as well as orthogroups (longest sequence from each orthogroup), were annotated with the following databases: carbohydrate-active enzymes (CAZy, downloaded 24 September 2021 [44]), peptidases (MER, MEROPS Scan Sequences, release 12.1 [45]), ABC transporters (ABC), transcription factors (TFs), cytochrome p450 oxidases (CYP), peroxidases (PER, downloaded 22 June 2022 [46]), host-pathogen interacting genes (PHI, downloaded 22 June 2022 [47]), as well as KOG genes (release 2003 [48]), and Pfam domains (v35 [49]). KOGs, peptidases, peroxidases, and PHI were searched with blastp using diamond v2.0.9 [50] with options “more sensitive” and e-value of 1 × 10^−10^. CAZy database was searched using diamond v2.0.9 with options “more sensitive” and e-value of 1 × 10^−102^. Pfam domains were searched with pfam_scan.pl [49] script using hmmer v3.2.1 [51]. Transcription factors were determined based on Pfam domains corresponding to Sordariomycetes transcription factors retrieved from JGI Mycocosm database (9 December 2021 download). Cytochrome p450 oxidases were determined based on the presence of the Pfam domain PF00067, and ABC transporters were determined based on the presence of the Pfam domains PF00005 and/or PF00664. Gene ontology (GO) terms were assigned based on Pfam domains using a mapping file v13/06/2022. Finally, Secondary Metabolite Clusters (SMCs) and corresponding genes were identified from genome assemblies using antiSMASH v4.0.2 [52]. In downstream analyses, we considered the sum of genes annotated within each SM cluster, and not just the number of clusters, in order to make analyses of gene gains and losses comparable across gene classes.

### 2.3. Phylogeny

We reconstructed a maximum likelihood phylogeny using a concatenated set of 3723 conserved genes (Busco v3) [41]. Protein sequences were aligned with mafft v7.453 [53] with E-INS-i method (options “genafpair”, “ep 0” and “maxiterate 1000”) and trimmed with trimAL v1.4.rev22 [54] using option “automated1”. Protein alignments were subsequently concatenated to create a matrix with 1,809,291 sites in total and 1,193,141 parsimony informative sites. The model of substitution was selected for each gene using IQ-TREE v1.6.12 [55,56], using the BIC score. LG + I + G4 was selected as the most common best model across all genes. The tree was built with IQ-TREE v1.6.12 [57] with ultrafast bootstrap and 1000 replicates. The tree was time-scaled using r8s v1.81 [58], with the calibration point of 299 Mya set at the split of *Neurospora crassa* with *Chalara longipes*. The time of split was retrieved from TimeTree [59].

### 2.4. Ancestral State Reconstructions

The vascular wilt-causing trait was reconstructed using ace function from the R package ape [60], with the “ER” model. We used the scaled likelihood of causing wilt estimated for each node (between 0 and 1), to identify the origins of vascular wilts. Namely, we set the origin of wilt-causing species as the most recent common ancestor (MRCA) of wilt-causing species whose scaled likelihood of causing wilt was above 0.5. This led to the identification of seven independent origins of vascular wilts.

The ancestral abundance of gene classes was estimated for the ancestral nodes using the fastAnc function in R package phytools [61]. We estimated the ancestral abundances in the MRCA of each wilt group (ANC wilt, scaled likelihood of causing wilt > 0.5) and the non-wilt-causing ancestor of wilt groups (ANC non-wilt, scaled likelihood of causing wilt < 0.5). We also estimated the abundances for the extant sister non-wilt-causing species or the MRCAs of non-wilt-causing species that are sister clades to wilt groups (non-wilt). In Ophiostomatales, the non-wilt-causing ancestor of wilt groups 6 and 7 was the same, because of the lack of a close sister clade/species that would not be derived from a wilt-causing ancestor. Fold change in gene family/group was calculated relative to the ancestral non-wilt node, as (ANC wilt—ANC non-wilt)/ANC non-wilt, and relative to the sister non-wilt-causing species as (ANC wilt—non-wilt)/non-wilt.

### 2.5. Statistical Analyses

Principal component analysis was run using the prcomp function in R with options “center = TRUE” and “scale = TRUE”. The correlation between the vascular wilt trait and the abundance of each gene family/group was calculated with the phylogenetic logistic regression using phyloglm function implemented in R package phylolm [62]. The method applied was “logistic_MPLE” and the options used were “btol = 10”, “boot = 1000”, and “log.alpha.bound = 0.005”.

The diversification of orthogroups (gene families) on the tree was estimated with CAFE v5 [63]. We used orthology relationships obtained with Orthofinder and removed large orthogroups with more than 80 genes. First, we ran CAFE to estimate overall lambda and obtain orthogroup counts. We removed orthogroups that had no reliable count estimates at the root (count equal to 1) and reran CAFE on the filtered dataset to obtain final estimates of gene expansions and contractions on the tree. For the subsequent analyses, we considered only orthogroups with significant changes at *p* < 0.05. To estimate the enrichment of orthogroup expansions (or contractions) simultaneously in several wilt groups, we calculated the probability of an orthogroup expansion in 1 up to 7 wilt groups using simulations. We simulated 1000 orthogroups using lambda estimated in the second run (lambda = 0.003347) and the distribution of orthogroup counts at the root, also obtained in the second run. Then we calculated the fraction of all simulated orthogroups that were expanding and present in 1–7 wilt groups (precisely at the MRCAs of the 7 wilt groups). The enrichment of Gene Ontology (GO) terms and Pfam domains was estimated by calculating the probability of finding the respective term as many times as there were wilt groups (the MRCAs of the wilt groups) actually annotated with the term. For this purpose, we reshuffled IDs of expanding (or contracting) orthogroups across tree nodes 1000 times, by retaining the exact number of orthogroups per node, and making sure that all orthogroups within each node were unique, and then counted the frequency of each term in all wilt groups. Terms with a low probability of occurrence (less than 0.05) in as many wilt groups as there in reality were, were considered significantly enriched.

Orthologs of the wilt-unique set of genes found in [27] were searched in all species gene models (protein sequences) using Orthofinder v2.5.2 [43], for protein sequences of all 76 species and protein sequences of *V. dahliae* VdLs.17 (Gene Bank assembly ASM15067v2).

All code was run using R v4.2.1 and Python v3.9.

## 3. Results

### 3.1. Wilt Pathogens Evolved Multiple Times

We reconstructed a maximum likelihood phylogeny of 73 Ascomycete species from Sordariomycetes class rooted in three species from the order Xylariales (Figure 1, Appendix A). This phylogeny encompassed 20 pathogens causing vascular wilt in plants, as well as their related pathogenic or saprotrophic species. Wilt species and strains are spread across the following four orders: Hypocreales (*Fusarium oxysporum* f. sp. *lycopersici* [*FOL*] causing tomato wilt, *F. oxysporum* f. sp. *cubense* causing Panama wilt disease of banana, and *F. xylarioides* causing wilt of coffee), Microascales (*Ceratocystis eucalypticola* causing eucalyptus wilt, *C. manginecans* causing mango wilt, *C. cacaofunesta* causing cacao wilt, *C. platani* causing canker stain in plane tree, and *Bretziella fagacearum* causing oak wilt), Glomerellales (*Verticillium albo-atrum, V. dahliae, V. klebahnii, V. tricorpus*, and *V. nubilum*, causing Verticillium wilts of annual and perennial plant species, among them important agricultural crops), and Ophiostomatales (*Ophiostoma novo-ulmi, O. ulmi*, and *O. himal-ulmi* causing Dutch elm disease [DED], *Leptographium procerum* and *L. wageneri* causing conifer wilts, *Raffaelea quercivora* causing Japanese oak wilt, and *R. lauricola* causing Laurel wilt).

To investigate how many times wilt species have evolved, we calculated the scaled likelihood of causing wilt for each node on the tree. By tracing back the ancestral tree nodes with a likelihood of causing wilt of 0.5 or above, we identified seven independent nodes that were the most recent wilt-causing ancestors of the current wilt-causing species (marked with red dots and numbers in the tree shown in Figure 1). *Fusarium* wilts (1), *Verticillium* wilts (4), and *Ophiostoma* wilts (5) all form monophyletic groups of wilt pathogens. *Bretziella fagacearum* (3) and *Raffaelea lauricola* (7) comprise single wilt pathogens, whereas *Ceratocystis* wilts (2), and *Raffaelea/Leptographium* (6) form paraphyletic groups of wilt pathogens, with subsequent loss of wilt-causing traits in some species.

The analysis showing that the ability to cause wilt has evolved multiple times (around seven times) across Sordariomycetes raises further hypotheses about parallel or distinct molecular mechanisms of host infection.

### 3.2. Pathogen-Related Gene Repertoires Do Not Distinguish Wilt from Non-Wilt-Causing Species

We compared the gene repertoires of several functional classes of genes to look for patterns of diversification specific to wilt-causing fungi (*n* = 20) among pathogenic species only (*n* = 39). We extracted the gene models of protein-coding genes from the assemblies of all wilt- and non-wilt-causing species and annotated them with a wide range of functions such as CAZymes (CAZy), secondary metabolite cluster genes (SMC), peptidases (MER), ABC transporters (ABC), transcription factors (TFs), cytochrome p450 oxidases (CYP), peroxidases (PER), and host-pathogen interacting genes (PHI).

The principal component analysis (PCA) based on the abundance of all gene classes roughly clusters the species belonging to four different orders according to the first three principal components, exposing a phylogenetic signal in the gene composition. Within orders, wilt-causing species are slightly separated from non-wilt-causing pathogenic species, in particular, according to PC2 and PC3. However, this pattern may be partly due to a phylogenetic effect, because in Glomerellales and Hypocreales wilt-causing fungi are monophyletic, whereas, within Ophiostomatales, only one non-wilt-causing species is present in our dataset (Figure 2). PC2 and PC3 place wilt-causing species from Hypocreales and Glomerellales close to each other, and similarly, those from Ophiostomatales and Microascales, but their clusters do not overlap. This indicates a phylogenetic impact on the composition of pathogen-related gene repertoires in wilt and non-wilt-causing fungi.

We further investigated if any pathogen-related gene class or function extracted from the KOG database is correlated with wilt-causing pathogens using phylogenetic logistic regression. None of the gene class abundances showed a significant correlation with vascular wilts (Appendix A). This suggests that molecular signatures of vascular wilts are either unique to individual taxonomic orders or they are driven not by broad gene classes but rather by a small subset of genes.

To see which gene classes are more abundant in wilt-causing species than in non-wilt-causing pathogenic species, we compared their median abundance in the wilt- and non-wilt-causing pathogens within each order (Figure 3A, Appendix A). *Fusarium* wilts (Hypocreales) showed, on average, a higher abundance of genes involved in defense mechanisms, when compared with several other plant pathogens of the same order (*Fusarium graminearum, F. poae, Cylindrodendrum hubeiense, Calonectria pauciramosa, Neonectria ditissima*, and *Pseudonectria buxi*). Wilt pathogens from the order Microascales had on average more CAZymes from the polysaccharide lyases (PL) family, although only by one gene on average, and only compared with two non-wilt-causing plant pathogens, *Ceratocystis adiposa*, and *Thielaviopsis musarum* (Appendix A). The wilt pathogens from Microascales, also showed, on average, fewer SMC genes from the PKS (polyketide synthase) and terpene family. In the order Glomerellales, three specific SMC families (PKS, NRPS—non-ribosomal peptide synthetase, and terpene) and cytochrome p450 oxidases were less abundant compared to the plant pathogenic *Plectosphaerella cucumerina* and five *Colletotrichum* species. In Ophiostomatales, all except for one pathogen (the human pathogenic *Sporothrix schenckii*) are causing wilt. Wilt pathogens had, on average, more CAZymes from the PL family, (as in Microascales, also only by one gene on average, Figure 3B), and SMC genes from the PKS family, while they carried fewer SMC genes from the NRPS and “other” families.

### 3.3. Secondary Metabolite Cluster (SMC) Diversification in Distinct Wilt Groups

Our phylogenetic reconstruction of the wilt-causing trait suggests at least seven independent events when vascular wilts evolved (hereafter wilt groups). This created opportunities for many distinct molecular mechanisms of vascular wilt to evolve. We investigated patterns of expansions and contractions of gene classes at the most recent common ancestor of wilt-causing species within each of the seven wilt groups. For this purpose, we inferred the abundance of each gene class in the most recent wilt-causing ancestor of each wilt group (red dots in Figure 1) and compared them with the abundance in the most recent non-wilt-causing ancestor (open dots in Figure 1). We also compared gene abundance in the most recent wilt-causing ancestor with the existing non-wilt-causing species (or the most recent ancestor of non-wilt-causing species, black dots in Figure 1).

*Fusarium* wilts showed a 21% increase in genes involved in defense mechanisms (V) compared to the non-wilt ancestral node (Figure 4A and Appendix A, Appendix A). *Verticillium* wilts showed a 28% increase in CAZymes from the polysaccharide lyases (PL) family and a 25% increase in genes belonging to “other” SMCs. Two wilt groups (*Bretziella fagacearum* and *Raffaelea lauricola*) showed an increase (19 and 22%, respectively) in SMC genes from the PKS (polyketide synthase) group, and two wilt groups (*Ophiostoma* wilts and *Raffaelea*/*Leptographium*) showed an increase in SMC genes from the terpene group, although only by 8% in *Ophiostoma* wilts, and 13% in *Raffaelea/Leptographium*. The *Raffaelea/Leptographium* wilt group also showed an 11% increase in genes involved in cell motility (N), and *R. lauricola* had an overall 24% increase in SMC genes. *B. fagacearum* and *R. lauricola*, similar to *Verticillium* wilts but unlike other Microascales and Ophiostomatales wilts, had more PL genes than the non-wilt ancestral node, although these differences were not significant (Appendix A). SMC families were also among the most reduced gene classes in every wilt group, except for *Ophiostoma* wilts.

When abundances were compared with a non-wilt-causing sister species or node, most wilt groups showed similar patterns of gene fold change, as observed when compared to the non-wilt ancestral node (Figure 4B and Appendix A, Appendix A). Most noticeable differences were found in *Verticillium* wilts, which showed an overall decrease in the abundance of SMCs including PKS and NRPS (non-ribosomal peptide synthetase) since the most recent ancestor. Still, the decrease was less pronounced than in the sister non-wilt-causing pathogen *Plectosphaerella cucumerina*, which had 80% fewer genes from the PKS cluster and 42% fewer genes from all SMCs than *Verticillium* wilts. The *Ophiostoma* wilts showed the highest fold change (8%) of genes from the terpene cluster since the most recent non-wilt-causing common ancestor but not when compared with the non-pathogenic *Ophiostoma quercus*. Instead, *Ophiostoma* wilts had 13% more genes from the NRPS SMC compared to *O. quercus*.

### 3.4. Gene Diversifications Shared among Wilt Groups

We searched for orthologs in all species (*n* = 76) and identified gene family expansions and contractions on the phylogenetic tree. We detected gene families that expanded or contracted preferentially in wilt groups, by simulating gene family expansions and contractions on the tree and calculating how likely family abundance can change in 1 up to 7 wilt ancestral nodes simultaneously. The four expanded gene families (orthogroups: OG0000010, OG0000011, OG0000017, and OG0000093) had a significantly higher occupancy across wilt groups than expected from simulations (in 4 out of 7, *p* = 0.01, Figure 5). Only wilt group 6 (*Raffaelea/Leptographium*) showed losses instead of gains in all four of those gene families (Figure 5). Only one gene family contraction (OG0000015, hypothetical protein) was enriched in wilt groups (in 4 out of 7, *p* = 0.016) and there were no gene family expansions or contractions unique to wilt-causing species.

OG0000010 matched quinate permease. In yeast, this gene family had the best match to the SEO1, a putative permease with allantoate transmembrane transporter activity, and FEN2, a plasma membrane H+-pantothenate symporter involved in endocytosis, with mutants showing reduced resistance to heat, chemicals, and hyperosmotic stress. Quinate was suggested as a major carbon source during plant host infection in *Magnaporthae oryzae* [64]. OG0000011 was annotated as a (thermostable) beta-glucosidase B, which catalyzes the hydrolysis of β-D-glucosyl residues to form β-D-glucose. In yeast, this protein matched STL1, the glycerol proton symporter of the plasma membrane induced during cell osmotic shock. Both gene families (OG0000010 and OG0000011) are involved in transmembrane transport (GO:0055085), and transmembrane transporter activity (GO:0022857), and are integral components of the membrane (GO:0016021). They both also have a sugar-transporting domain (PF00083), whereas OG0000011 was additionally annotated as CAZyme, from the glycoside hydrolase (GH) family, and OG0000010 had a major facilitator superfamily domain. OG0000017 and OG0000093 were annotated as hypothetical proteins, the first one containing a phosphotransferase enzyme family domain (PF01636) and the second one, a Pfam domain CHAT (PF12770).

*O. novo-ulmi* has 15 members in family OG0000010, 8 members in each family OG0000011 and OG0000017, and only one current member of family OG0000093. We checked the transcript expression using previously published experimental in vitro and in planta studies [65,66,67] and one unpublished results study (T. de Oliveira, unpublished). One member of family OG0000010 and two members of family OG0000017 had a high or medium expression level in vitro in both yeast and mycelium growth phases (Appendix A). Three members of the OG0000011 family showed medium to high expression during infection in planta, two of which also showed elevated expression in vitro, in particular during the yeast phase (Appendix A). We could not find the corresponding protein of OG0000093 among *O. novo-ulmi* H327 protein sequences, likely due to the discrepancies between our gene annotation and the former gene annotation of the *O. novo-ulmi* H327 genome.

We also checked for the enrichment of gene ontology (GO) functions and Pfam domains related to gene family expansions in ancestral nodes of wilt groups, by shuffling IDs of gene families across all tree nodes (retaining the frequency specific to each node) and calculating the probability of the presence of a given term in at least as many ancestral wilt groups as observed. Among functional annotations related to gene family expansions, two gene ontology terms enriched in wilt groups were carbohydrate metabolic process (GO:0005975, in 6 out of 7 wilt groups, all except in *Ceratocystis* wilts, *p* = 0.022), and an integral component of membrane (GO:0016021, all groups except *Raffaelea/Leptographium*, *p* = 0.048). The integral component of membrane term was also present among family contractions in six wilt groups (all except *Fusarium* wilts, *p* = 0.061). Four Pfam domains most commonly found among expanding gene families in wilt groups (*n* = 5) were sugar transporter (PF00083), acyl transferase domain (PF00698), cytochrome p450 domain (PF00067), and ankyrin repeat domain (PF12796), but none of them were significantly enriched (respective *p* values: 0.133, 0.146, 0.200, and 0.351). One Pfam domain CHAT (PF12770) was present among gene family contractions in 4 wilt groups (*p* = 0.031).

A previous study [27] found a bacterial glucosyltransferase in *FOL*, *V. dahliae,* and *V. albo-atrum*. By blasting the bacterial glucosyltransferase (ZP_01227727.1) against all gene orthologs, we found it in all *Fusarium* wilts, and *V. dahliae*, but not in other wilt species. We also searched for orthologs of 10 genes of *V. dahliae* that were found to be shared with *Fusarium* wilts but no other *Fusarium* species [27]. We found one of these to be shared by Ophiostomatales and Microascales wilts, but at the same time, they were also shared with other non-wilt species (Appendix A). Similar to [27], we checked for genes unique to wilt-causing species in our dataset. Specifically, we wanted to see if any gene homologs are present in at least two independent wilt groups but are not shared with other species. None of the gene homologs were found in all wilt groups. We found 28 genes shared by at most 2 independent wilt groups, with *Fusarium* and *Verticillium* wilts sharing 11 genes (including glucosyltransferase), *Raffaelea/Leptographium* and *R. lauricola* wilts sharing 7, *Fusarium* and *R. lauricola* wilt sharing 3, *Verticillium* and *Ophiostoma* wilts sharing 2, and the rest of the pairs sharing one gene (Appendix A). Only six of these genes had a match to a described protein, and they included some genes potentially important for virulence, such as genes encoding MFS-type transporter *prx5*, or ABC-type transporters. Apart from glucosyltransferase, agropine synthesis cyclase was another gene potentially transferred horizontally from bacteria and shared between *Fusarium* and *Verticillium* wilts.

Inspection of *Ophiostoma* transcript expression for gene families unique to wilt groups revealed one gene family (OG0016786) with a high expression level in vitro in the mycelium stage, as well as a modest expression level in planta (Appendix A). Another gene family (OG0019922) showed a similar pattern with an additional significantly differential expression between yeast and mycelium phase (Appendix A). Although this gene family was shared only by *O. ulmi* and *L. procerum* species according to our analysis, it had a blast hit to an *O. novo-ulmi* H327 protein in the JGI Mycocosm database, suggesting that either this may not be a strict ortholog of the *O. ulmi* gene, or that our gene model inference in *O. novo-ulmi* did not capture this gene.

### 3.5. Dutch Elm Disease

Species from the *Ophiostoma* wilt group have an estimated number of protein-coding genes between 8413 and 8453; this is over 400 fewer genes than in their sister non-pathogenic *O. quercus*, although not the lowest number found within Ophiostomatales (Appendix A). Most of the gene classes have fewer genes in *Ophiostoma* wilts than in *O. quercus* (Appendix A), and most significant losses are seen in CAZymes, such as CBM (48 on average in DED compared to 60 in *O. quercus*), AA (46 on average in DED compared to 61 in *O. quercus*), and PL families (3 in *O. himal-ulmi* compared to 4 in *O. quercus*), as well as in SMCs from the “other” family (34 on average in DED compared to 41 in *O. quercus*). The only two gene classes analyzed by us that undergo fold increase with Z-score above 2 in *Ophiostoma* wilts include SMCs from the NRPS family (10 on average in DED compared to 9 in *O. quercus*), and terpenes (22 in *O. novo-ulmi* compared to 20 in *O. quercus*). Another two gene classes with non-significant fold increases include only SMCs from the PKS family (103 on average in *Ophiostoma* wilts compared to 99 in *O. quercus*), and cytochrome p450 oxidases (46 on average in *Ophiostoma* wilts compared to 44 in *O. quercus*). Even though these are very modest gains, it is important to keep in mind that except for these four gene classes, all other classes were shown to reduce their numbers. We also found that *Ophiostoma* wilt pathogens and their immediate ancestors have in total 22 gene families that undergo expansions vs. as many as 33 gene families that undergo contractions (10 of which are in common). In contrast, *O. quercus* has 29 gene family expansions (the biggest expansion being in the gene family OG0000017), and only one gene family contraction (gene family OG0000010). The biggest contractions in DED fungi are in the gene family OG0000017 (total loss of 5–7 genes since the most recent common ancestor of *Ophiostoma* wilts). Other gene families with the highest gene gains and losses include one gene family expansion and one gene family contraction, both of which are annotated as having the transferring phosphorus-containing groups and protein kinase-like domains. Our ortholog search revealed about 189 gene families unique to all three *Ophiostoma* wilt pathogens, 29 unique to *O. novo-ulmi*, 31 to *O. ulmi,* and 44 to *O. himal-ulmi*, suggesting some possibilities of adaptation to the host via novel or horizontally transferred genes.

## 4. Discussion

### 4.1. Genomics of Wilt Pathogens

Fungal wilt pathogens are responsible for several pandemics that have resulted in heavy financial losses annually and, in the case of forest tree species, have devastated populations of susceptible hosts and led to landscape changes. In the case of Dutch elm disease (DED), which was first described in the early 1920s [8], the pathogens *O. ulmi* and *O. novo-ulmi* caused two successive pandemics that impacted several of the elm species in the forest, agrarian, and urban environments on several continents [68,69,70]. Within 100 years, an estimated 1 billion adult elm trees were killed [71], and the ongoing pandemic caused by *O. novo-ulmi* is threatening the survival of the iconic species *Zelkova carpinifolia* [72].

The sequencing and annotation of genomes of *O. ulmi* [73] and *O. novo-ulmi* [38,74] opened the way for transcriptomic approaches to complement previous strategies used for unraveling the molecular bases of various biological traits in the DED fungi. Thus, the genes and pathways associated with yeast-mycelium dimorphism in vitro [65,66] and pathogenicity in planta [67] were identified and are now being validated by loss-of-function assays with targeted deletion mutants [75]. The wilt-causing fungi have so far been recorded in orders Hypocreales, Microascales, Glomerellales, and Ophiostomatales within the class Sordariomycetes. Therefore, whole-genome comparisons that may identify signature genes or genomic regions that are specific to wilt pathogens complement the above-mentioned approaches. Several candidate genes have been proposed from the analyses of six species in the genera *Fusarium* and *Verticillium* for which sequenced genomes were available [26,27]. Since this time more fungal genomes have been sequenced, and we were able to carry out a more exhaustive comparison of 20 wilt-causing species, along with 57 phylogenetically close species that included other types of pathogens as well as saprotrophs. A central question we wanted to address was whether the ability to cause wilt resulted from parallel or distinct genomic changes. A parallel evolution scenario was expected to yield a clear molecular signature for the ability to cause wilt that would span phylogenetically diverse organisms. For instance, the genomes of all wilt-causing pathogens would have the 10 genes (or at least a subset of these genes) shared by *V. albo-atrum*, *V. dahliae,* and *FOL* [27]. The alternative scenario, in which the ability to cause wilt was acquired through independent events during evolution, was expected to show similarities between phylogenetically related wilt-causing species but not between phylogenetically distant ones.

### 4.2. Differences in Gene Repertoires across Wilt Pathogens

The abundance of gene classes related to the pathogenicity is strongly dependent on the taxonomic order to which the wilt pathogen belongs, although some similarities can be drawn between *Fusarium* and *Verticillium* wilts, and between Ophiostomatales and Microascales wilts. These similarities are most likely a consequence of the distinct genome evolution patterns in these groups [40]. Fungi in Hypocreales and Glomerellales (in particular, plant pathogens) tend to have medium- or large-sized genomes (average 44.3 and 43.3 Mbp in the two orders, respectively) and a large number of genes (average 11,377 and 10,657 in the two orders, respectively). Ophiostomatales, and most prominently Microascales, show an opposite pattern with small- and medium-sized genomes (average 31.2 and 30.7 Mbp in the two orders, respectively), and a smaller number of genes (average 9115 and 7333 in the two orders, respectively). *Fusarium* and Glomerellales fungi exhibit gene diversification in most of the pathogen-relevant gene classes (e.g., secondary metabolite clusters [SMCs], carbohydrate-active enzymes [CAZymes], transcription factors), whereas Microascales and Ophiostomatales experience losses in those genes [40]. As it has been previously suggested [74], such patterns of reductive evolution could be linked to the dependence of Microascales and Ophiostomatales fungi on insect transmission, and the dependence of fungi on insect abilities to penetrate host tissues. In effect, gene repertoires are not directly comparable between pathogens from these four taxonomic orders.

Our genome comparison within taxonomic orders revealed differences in the evolution of gene families and functional gene groups among different wilt pathogens and no common signatures differentiating wilt and other types of pathogens. It is not unexpected given that wilt species from our dataset have a wide spectrum of hosts and occupy various niches. Pathogens, in general, evolve in environments with different sets of biotic and abiotic conditions; therefore, abundance changes in functional gene groups are expected to better reflect some specific lifestyle-related traits of pathogens (e.g., losses and gains of plant and microbe cell wall degrading enzymes in mycorrhizal fungi [76]), rather than the ability to induce wilt. Moreover, our detection of outlier gene classes is contingent on the choice and spectrum of pathogens in the control (non-wilt) group of pathogens. For example, Glomerellales fungi compared to *Verticillium* wilts, are dominated by a single genus, *Colletotrichum*, which itself is characterized by high numbers of CAZymes, SMC genes, and small secreted proteins [32,77]. In Ophiostomatales, non-wilt-causing pathogens are scarce, and in our dataset represented by a human pathogen *Sporothrix schenckii*, an exception among mostly tree-dwelling fungi. This highlights the importance of sequencing more fungal species from under-represented ecological groups to draw better conclusions regarding the evolution of fungal pathogens. The most convincing example of gene diversification might be the higher number of genes involved in defense mechanisms in *Fusarium* wilts (an average of 76 compared to 59 in non-wilt species). This result was consistent in comparison of *Fusarium* wilts to all pathogens, their sister group of non-wilt *Fusarium* pathogens, and non-wilt most recent common ancestor.

### 4.3. Common Patterns of Gene Diversification between Wilt Groups

We hypothesized that the gene changes most relevant for developing wilt-inducing characteristics would be best observed in the most recent common ancestors of each independent wilt-inducing group of species. By inferring ancestral states of gene class abundances and inferring expansions and contractions of gene homologs in those ancestral nodes, we found that a few gene class expansions are shared by some, but not all of the wilt group ancestors. Interestingly, some similarities are shared among taxonomic orders, but not always among wilt groups from the same order. Specifically, we observed the common fold increases or decreases in genes encoding SMCs. SMCs such as PKS (polyketide synthases) tend to diversify in *B. fagacearum* and *R. lauricola* wilt, whereas terpenes diversify in *Ophiostoma* wilts and *Raffaelea/Leptographium*. This result confirms the observation of Ibarra Caballero et al. [78] who compared eight species of Ophiostomatales and found more PKS clusters in *R. lauricola* than in the non-pathogenic *R. aguacate*, with nine PKS clusters unique to two strains of *R. lauricola*. On the contrary, some SMC gene families decrease in number across different wilt groups. This high turnover of genes in SMCs may suggest the importance of secondary metabolism in wilt pathogens.

Secondary metabolite clusters have a wide range of actions and functions (reviewed in [79,80]), including mediation of virulence on the host, production of UV-protective melanin, regulation of development, antibacterial or antifungal activity, or protection against insects. One potential explanation for the significant fold increases and decreases of genes within SMCs, especially in Microascales and Ophiostomatales, is that the generally high turnover of genes involved in the synthesis of secondary metabolites may underlie fungus-insect mutualistic relationships. Synthesized metabolites may be used either to attract insect vectors, as in the case of *Drosophila* being attracted to volatile-producing yeast [81], or can be used to mitigate the impact of plant-produced volatiles that are detrimental to insects, such as in the case of black fungi grown in the ant-plant symbiotic systems [82].

The same pattern is not observed however in other pathogenicity-related genes. For example, *Verticillium* wilts show diversification of CAZymes from the PL (polysaccharide lyases) family (on average 39 compared to 30 in the non-wilt ancestor). From other wilt groups, only *B. fagacearum* shows an overall fold increase in genes in this group (seven compared to six in the non-wilt ancestor and four in the sister pathogen), whereas the other Microascales and Ophiostomatales wilt groups experience losses or no change in PL number. Ophiostomatales tend to lose CAZymes, including CBM (carbohydrate-binding modules) in DED (an average of 48 compared to 52 in non-wilt ancestor, and 60 in *O. quercus*), and CE (carbohydrate esterases) in *Raffaelea/Leptographium* or *R. lauricola* wilt (on average 21 or 27 compared to 33 in non-wilt ancestor and 45 in *R. ambrosiae*).

Although we did not observe outstanding fold change in genes annotated with KOG carbohydrate transport and metabolism function in any of the wilt groups, interestingly, we found in wilt groups an enrichment of four gene family expansions, two of which are involved in sugar transport and carbohydrate metabolism. Moreover, the GO term carbohydrate metabolic process was enriched among gene expansions among wilt group ancestors. While looking for any gene homologs uniquely shared by wilt groups, at least four of them were also involved in some kind of carbohydrate transport or metabolism. This suggests that even though the overall balance of expansions and contractions of genes involved in carbohydrate metabolism is close to 0, some particular gene expansions may be beneficial specifically to wilt pathogens. A previous study found horizontally transferred bacterial glucosyltransferase shared by *Fusarium* and *Verticillium* wilts, involved in osmoregulation in bacteria [27]. Our results would confirm that mechanisms of osmoregulation required in the sugar-poor environment of the xylem may underlie the duplications among certain genes involved in sugar transport and metabolism. We further highlight an efflux pump of the ATP-binding cassette (ABC) transporter superfamily among genes shared between representatives of different pairs of wilt groups, involving highly virulent (aggressive) DED strains. While future functional studies will more precisely elucidate this gene’s involvement in the arms race between DED and its main host, fungal *PDR* (*Pleiotropic Drug Resistance*) genes are generally known for their rapid adaptive evolution and the formation of paralogs, and thus, ABC transporter proteins of secondary metabolites may contribute to a strain’s invasiveness [83,84].

### 4.4. No Large-Scale Expansions in DED Pathogens

Unlike many plant pathogens, DED pathogens tend to lose genes instead of gaining them when compared with the *O. quercus* sister species. Gene family expansions are accompanied by gene family contractions, leading to net gene losses in the genome. The classes of genes that appear to be least affected by the overall genomic contractions are SMCs, in particular those belonging to the terpene family. Therefore, the high virulence of DED may instead be more likely driven by a few large-impact genes and occurs via molecular evolution of existing genes, or acquisition of novel genes through duplications, de novo, or via horizontal gene transfer.

Our previous experimental studies [65,66,67] and unpublished results allowed us to examine some of the gene families unique or enriched in wilt species that we found in this study, by looking at their expression in *Ophiostoma novo-ulmi* and other DED species after in vitro or in planta infections. A few of them proved to be good candidates for causing virulence in elms, by showing elevated expression in planta after infection and represent potential candidates for future loss-of-function studies.

## 5. Conclusions

Here we present the genomic comparison of 76 species from the class Sordariomycetes to infer shared and distinct patterns of gene duplication among 20 pathogens causing vascular wilts in plants. *Fusarium* and *Verticillium* wilts evolve through the diversification of genes involved in defense mechanisms and through the diversification of carbohydrate-active enzymes, respectively. In Microascales and Ophiostomatales wilt groups, the most affected gene classes are different families of secondary metabolite clusters. In spite of largely distinct patterns of evolution in independent wilt groups, some gene family expansions involved in carbohydrate metabolism are shared among many of them, and they comprise a promising set of candidates for conferring virulence and promoting vascular wilt in infected plants.

## Figures and Tables

**Figure 1 jof-09-00002-f001:**
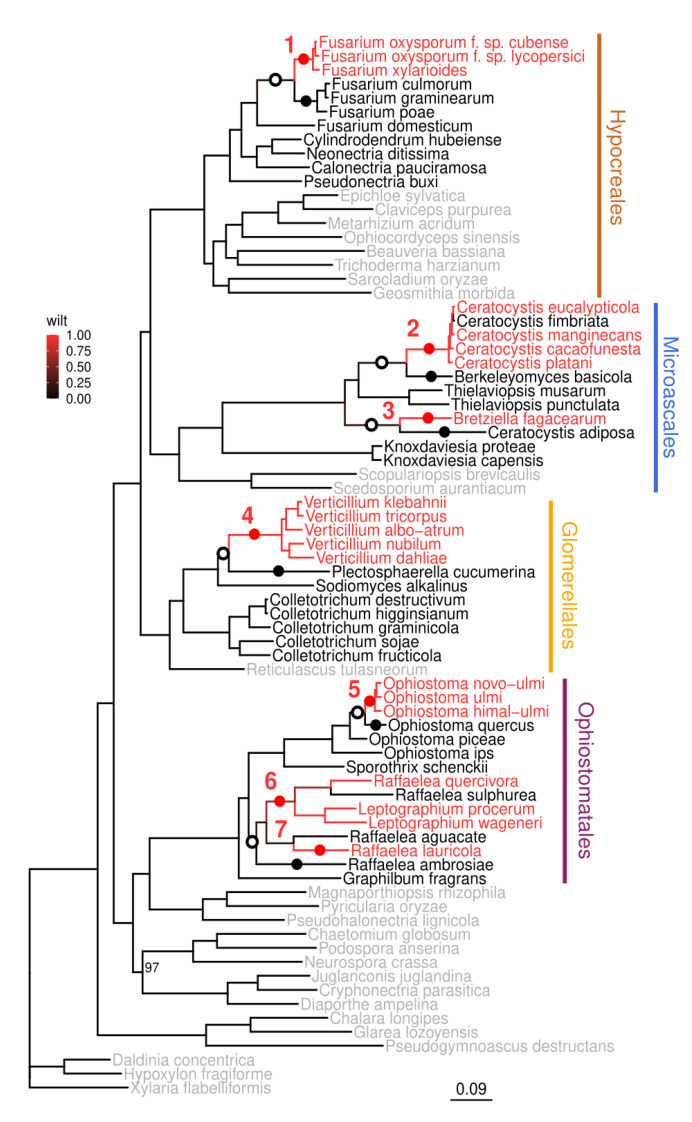
Seven origins of wilt-causing pathogens in Sordariomycetes. Maximum phylogeny in Sordariomycetes based on 3723 conserved single-copy genes calculated with IQ-TREE. Bootstrap support for all nodes is 100, except for one (97) shown on the tree. Species causing wilt are marked with red, species not causing wilt with are marked with black, and species not causing wilt and specifically chosen to support the phylogeny are marked with gray. Branch color corresponds to the scaled likelihood of causing wilt, estimated for all ancestral nodes using ace function from R package ape. Red dots indicate the most recent wilt-causing common ancestor of wilt-causing pathogens, black dots indicate non-wilt-causing species, or the most recent common ancestors of non-wilt-causing species used for comparison, open dots indicate the most recent common ancestor of wilt-causing species inferred as non-wilt-causing species, also used for comparison. Numbers indicate seven most likely transitions from non-wilt to wilt-causing pathogens.

**Figure 2 jof-09-00002-f002:**
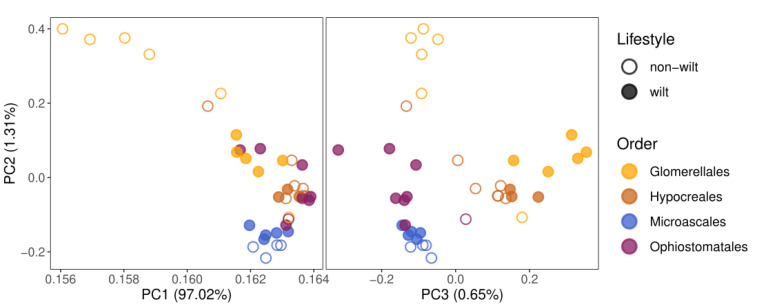
Principal component analysis, based on 17 pathogen-relevant gene class abundances, clusters species according to taxonomic order. Only pathogenic species were analyzed and included species, causing (filled dots) or not causing wilt (empty dots) diseases. Parentheses show the percent of explained variance.

**Figure 3 jof-09-00002-f003:**
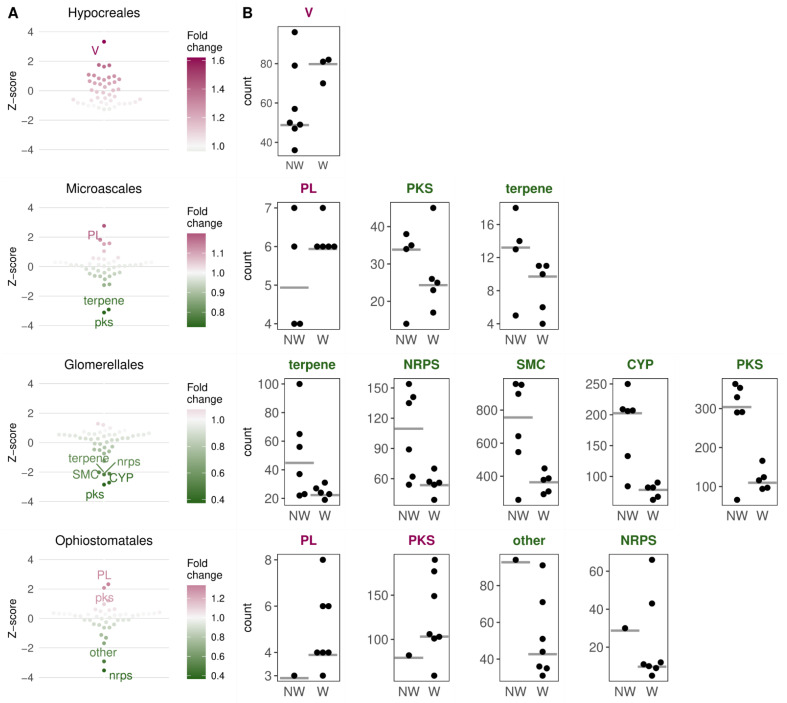
The most and least abundant gene classes in wilt relative to non-wilt-causing pathogenic species in four taxonomic groups of Sordariomycetes. (**A**) Normalized fold changes (z-scores) or ratios of median gene abundances in wilt vs. non-wilt-causing pathogens in four orders. The color indicates the strength of the fold change in wilt-causing relative to non-wilt-causing species. (**B**) Distribution of gene counts in gene classes whose medians (grey lines) were most differentiated between wilt- and non-wilt-causing fungi. Dots indicate gene counts of analyzed pathogenic species. NW—non-wilt-causing species, W—wilt-causing species. V—defense mechanisms (KOG function); PL—polysaccharide lyases; PKS—polyketide synthases; NRPS—non-ribosomal peptide synthetases; SMC—secondary metabolite clusters; CYP—cytochrome p450 oxidases; PKS, NRPS, terpene, “other” are all SMC families; PL is a CAZyme (carbohydrate-active enzyme) family.

**Figure 4 jof-09-00002-f004:**
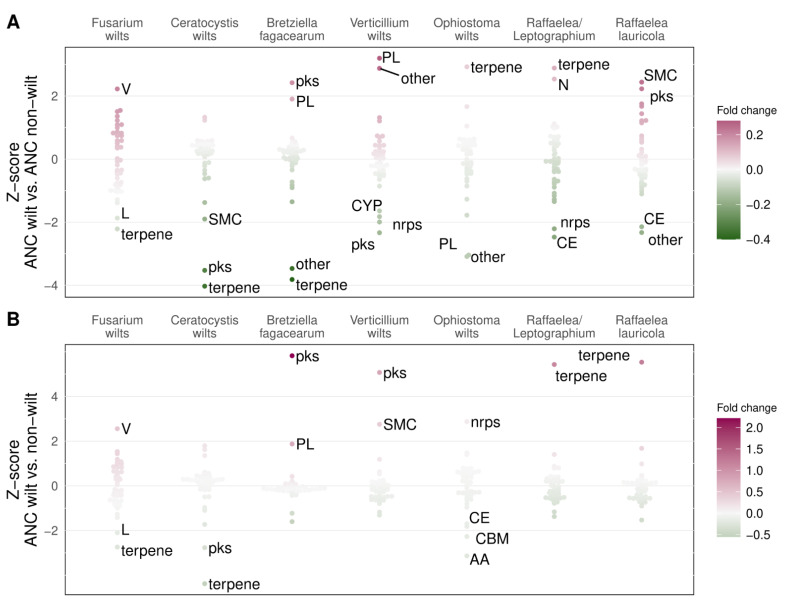
Normalized gene count ratios for gene classes in wilt-causing species (ANC wilt) vs. their non-wilt-causing most recent common ancestors (ANC non-wilt, (**A**)) or sister non-wilt species (non-wilt, (**B**)) in seven wilt groups. V—defense mechanisms; L—replication, recombination, and repair; N—cell motility; PL—polysaccharide lyases; CE—carbohydrate esterases; CBM—carbohydrate-binding modules; AA—auxiliary activities; pks—polyketide synthases; nrps—non-ribosomal peptide synthetases; SMC—secondary metabolite clusters; CYP—cytochrome p450 oxidases; V, L, and N are KOG functions; PKS, NRPS, terpene, “other” are all SMC families; PL, CE, CBM, and AA are all CAZyme (carbohydrate-active enzyme) families.

**Figure 5 jof-09-00002-f005:**
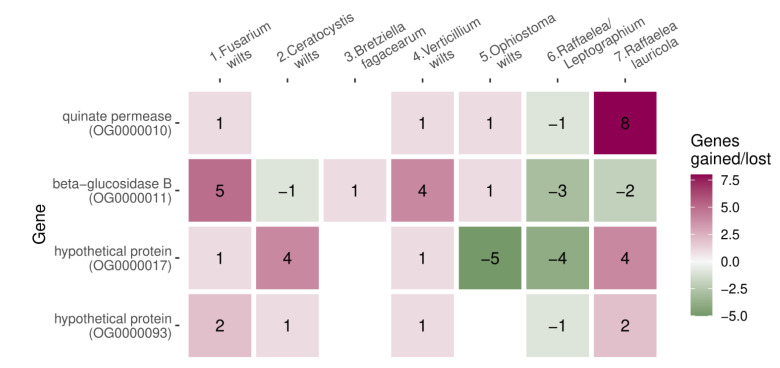
Gene families that diversified simultaneously in ancestral nodes of four wilt groups. Numbers indicate the number of gained or lost genes in the ancestor.

## Data Availability

Genome assemblies were retrieved from the public databases and their respective accession IDs, or links are listed in Appendix A.

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
