# Peer review of "Independent Evolution Has Led to Distinct Genomic Signatures in Dutch Elm Disease-Causing Fungi and Other Vascular Wilts-Causing Fungal Pathogens"

_jof, 2022, doi:10.3390/jof9010002_

Round 1
Reviewer 1 Report
This article provides significant scientific information on “Independent evolution has led to distinct genomic signatures in Dutch Elm Disease-causing fungi and other vascular wilts- 3 causing fungal pathogens.” No doubt that “Vascular wilts” is the serious issue in various parts of the world. The authors analyzed 20 whole genome assemblies of wilt causing fungi together with 56 other species using phylogenetic approaches to trace expansions and contractions of orthologous gene families and gene classes related to pathogenicity. They found that wilt-causing pathogens evolved seven times, experiencing the largest fold changes in different classes of genes almost every time. The results depicted that wilt-causing species evolve mostly through distinct changes in their repertoires of pathogenicity-related genes, and that there is the potential importance of carbohydrate metabolism genes for regulating osmosis in those pathogens that penetrate the plant vascular system. The finding are scientifically sound and important.
The “Introduction& materials and methods,” sections are described well and contains satisfactory information.
Results & Discussion depicted that Fusarium and Verticillium wilts evolve through diversification of genes involved in defense mechanisms and through diversification of CAZymes. In Microascales and Ophiostomatales wilt groups, the most affected gene clas-716 ses are different families of SMCs. In spite of largely distinct patterns of evolution in in-717 dependent wilt groups, some gene family expansions involved in carbohydrate metabo-718 lism are shared among many of them, and they comprise a promising set of candidates 719 for conferring virulence and promoting vascular wilt in infected plants. The statistical analysis performed are satisfactory with appropriate methodology. The results obtained are reliable.
Reviewer 2 Report
This manuscript presents a large body of interesting information from the multi-genomic analysis of wilt-causing pathogens in Sordariomycetes. The analysis unravels seven origins of fungal pathogens that cause not only wilt diseases of agricultural crops and economic plants but also Oak wilt or Dutch elm disease worldwide at the first time. The study is well done and all data are properly organized and presented. Also, the manuscript is well written. I believe the manuscript to benefit all plant wilt researchers and recommend it to be accepted for publication in JoF after a minor revision.
Minor suggestions for authors' consideration:
1. Subsection 2.6 for a list of abbreviations is unnecessary. Since most abbreviations are present in Figures 3 and 4, it would be better to see them in the figure legends. This modification would make the figures more self-understandable.
2. Following the journal's format, each figure with its legends must be inserted into the first paragraph of text where it is first cited. Pay attention to keeping each figure and its legends on the same page by adjusting figure size or length.
3. Repeat proof reading prior to resubmission is needed to minimize typos or make it for complicated phrases to be readily understood.
4. Check up the whole list of references in a right format for JoF and whether all scientific names in References are italicized.
